# Generative Adversarial Nets

**Ian J. Goodfellow**,* **Jean Pouget-Abadie**,† **Mehdi Mirza, Bing Xu, David Warde-Farley,**
**Sherjil Ozair**,‡ **Aaron Courville, Yoshua Bengio**§
Département d'informatique et de recherche opérationnelle
Université de Montréal
Montréal, QC H3C 3J7

## Abstract

We propose a new framework for estimating generative models via an adversarial process, in which we simultaneously train two models: a generative model $G$ that captures the data distribution, and a discriminative model $D$ that estimates the probability that a sample came from the training data rather than $G$. The training procedure for $G$ is to maximize the probability of $D$ making a mistake. This framework corresponds to a minimax two-player game. In the space of arbitrary functions $G$ and $D$, a unique solution exists, with $G$ recovering the training data distribution and $D$ equal to $\frac{1}{2}$ everywhere. In the case where $G$ and $D$ are defined by multilayer perceptrons, the entire system can be trained with backpropagation. There is no need for any Markov chains or unrolled approximate inference networks during either training or generation of samples. Experiments demonstrate the potential of the framework through qualitative and quantitative evaluation of the generated samples.

## 1   Introduction

The promise of deep learning is to discover rich, hierarchical models [2] that represent probability distributions over the kinds of data encountered in artificial intelligence applications, such as natural images, audio waveforms containing speech, and symbols in natural language corpora. So far, the most striking successes in deep learning have involved discriminative models, usually those that map a high-dimensional, rich sensory input to a class label [14, 20]. These striking successes have primarily been based on the backpropagation and dropout algorithms, using piecewise linear units [17, 8, 9] which have a particularly well-behaved gradient . Deep *generative* models have had less of an impact, due to the difficulty of approximating many intractable probabilistic computations that arise in maximum likelihood estimation and related strategies, and due to difficulty of leveraging the benefits of piecewise linear units in the generative context. We propose a new generative model estimation procedure that sidesteps these difficulties. [1]

In the proposed *adversarial nets* framework, the generative model is pitted against an adversary: a discriminative model that learns to determine whether a sample is from the model distribution or the data distribution. The generative model can be thought of as analogous to a team of counterfeiters, trying to produce fake currency and use it without detection, while the discriminative model is analogous to the police, trying to detect the counterfeit currency. Competition in this game drives both teams to improve their methods until the counterfeits are indistiguishable from the genuine articles.

This framework can yield specific training algorithms for many kinds of model and optimization algorithm. In this article, we explore the special case when the generative model generates samples by passing random noise through a multilayer perceptron, and the discriminative model is also a multilayer perceptron. We refer to this special case as *adversarial nets*. In this case, we can train both models using only the highly successful backpropagation and dropout algorithms [16] and sample from the generative model using only forward propagation. No approximate inference or Markov chains are necessary.

## 2   Related work

Until recently, most work on deep generative models focused on models that provided a parametric specification of a probability distribution function. The model can then be trained by maximizing the log likelihood. In this family of model, perhaps the most succesful is the deep Boltzmann machine [25]. Such models generally have intractable likelihood functions and therefore require numerous approximations to the likelihood gradient. These difficulties motivated the development of "generative machines"–models that do not explicitly represent the likelihood, yet are able to generate samples from the desired distribution. Generative stochastic networks [4] are an example of a generative machine that can be trained with exact backpropagation rather than the numerous approximations required for Boltzmann machines. This work extends the idea of a generative machine by eliminating the Markov chains used in generative stochastic networks.

Our work backpropagates derivatives through generative processes by using the observation that

$$\lim_{\sigma \to 0} \nabla_{\boldsymbol{x}} \mathbb{E}_{\epsilon \sim \mathcal{N}(0, \sigma^2 \boldsymbol{I})} f(\boldsymbol{x} + \epsilon) = \nabla_{\boldsymbol{x}} f(\boldsymbol{x}).$$

We were unaware at the time we developed this work that  Kingma and Welling [18] and  Rezende *et al.* [23] had developed more general stochastic backpropagation rules, allowing one to backpropagate through Gaussian distributions with finite variance, and to backpropagate to the covariance parameter as well as the mean. These backpropagation rules could allow one to learn the conditional variance of the generator, which we treated as a hyperparameter in this work.  Kingma and Welling [18] and  Rezende *et al.* [23] use stochastic backpropagation to train variational autoencoders (VAEs). Like generative adversarial networks, variational autoencoders pair a differentiable generator network with a second neural network. Unlike generative adversarial networks, the second network in a VAE is a recognition model that performs approximate inference. GANs require differentiation through the visible units, and thus cannot model discrete data, while VAEs require differentiation through the hidden units, and thus cannot have discrete latent variables. Other VAE-like approaches exist [12, 22] but are less closely related to our method.

Previous work has also taken the approach of using a discriminative criterion to train a generative model [29, 13]. These approaches use criteria that are intractable for deep generative models. These methods are difficult even to approximate for deep models because they involve ratios of probabilities which cannot be approximated using variational approximations that lower bound the probability. Noise-contrastive estimation (NCE) [13] involves training a generative model by learning the weights that make the model useful for discriminating data from a fixed noise distribution. Using a previously trained model as the noise distribution allows training a sequence of models of increasing quality. This can be seen as an informal competition mechanism similar in spirit to the formal competition used in the adversarial networks game. The key limitation of NCE is that its "discriminator" is defined by the ratio of the probability densities of the noise distribution and the model distribution, and thus requires the ability to evaluate and backpropagate through both densities.

Some previous work has used the general concept of having two neural networks compete. The most relevant work is predictability minimization [26]. In predictability minimization, each hidden unit in a neural network is trained to be different from the output of a second network, which predicts the value of that hidden unit given the value of all of the other hidden units. This work differs from predictability minimization in three important ways: 1) in this work, the competition between the networks is the sole training criterion, and is sufficient on its own to train the network. Predictability minimization is only a regularizer that encourages the hidden units of a neural network to be statistically independent while they accomplish some other task; it is not a primary training criterion. 2) The nature of the competition is different. In predictability minimization, two networks' outputs are compared, with one network trying to make the outputs similar and the other trying to make the

outputs different. The output in question is a single scalar. In GANs, one network produces a rich, high dimensional vector that is used as the input to another network, and attempts to choose an input that the other network does not know how to process. 3) The specification of the learning process is different. Predictability minimization is described as an optimization problem with an objective function to be minimized, and learning approaches the minimum of the objective function. GANs are based on a minimax game rather than an optimization problem, and have a value function that one agent seeks to maximize and the other seeks to minimize. The game terminates at a saddle point that is a minimum with respect to one player's strategy and a maximum with respect to the other player's strategy.

Generative adversarial networks has been sometimes confused with the related concept of "adversarial examples" [28]. Adversarial examples are examples found by using gradient-based optimization directly on the input to a classification network, in order to find examples that are similar to the data yet misclassified. This is different from the present work because adversarial examples are not a mechanism for training a generative model. Instead, adversarial examples are primarily an analysis tool for showing that neural networks behave in intriguing ways, often confidently classifying two images differently with high confidence even though the difference between them is imperceptible to a human observer. The existence of such adversarial examples does suggest that generative adversarial network training could be inefficient, because they show that it is possible to make modern discriminative networks confidently recognize a class without emulating any of the human-perceptible attributes of that class.

## 3 Adversarial nets

The adversarial modeling framework is most straightforward to apply when the models are both multilayer perceptrons. To learn the generator's distribution $p_g$ over data $\boldsymbol{x}$, we define a prior on input noise variables $p_{\boldsymbol{z}}(\boldsymbol{z})$, then represent a mapping to data space as $G(\boldsymbol{z}; \theta_g)$, where $G$ is a differentiable function represented by a multilayer perceptron with parameters $\theta_g$. We also define a second multilayer perceptron $D(\boldsymbol{x}; \theta_d)$ that outputs a single scalar. $D(\boldsymbol{x})$ represents the probability that $\boldsymbol{x}$ came from the data rather than $p_g$. We train $D$ to maximize the probability of assigning the correct label to both training examples and samples from $G$. We simultaneously train $G$ to minimize $\log(1 - D(G(\boldsymbol{z})))$. In other words, $D$ and $G$ play the following two-player minimax game with value function $V(G, D)$:

$$\min_G \max_D V(D, G) = \mathbb{E}_{\boldsymbol{x} \sim p_{\text{data}}(\boldsymbol{x})}[\log D(\boldsymbol{x})] + \mathbb{E}_{\boldsymbol{z} \sim p_{\boldsymbol{z}}(\boldsymbol{z})}[\log(1 - D(G(\boldsymbol{z})))]. \tag{1}$$

In the next section, we present a theoretical analysis of adversarial nets, essentially showing that the training criterion allows one to recover the data generating distribution as $G$ and $D$ are given enough capacity, i.e., in the non-parametric limit. See Figure 1 for a less formal, more pedagogical explanation of the approach. In practice, we must implement the game using an iterative, numerical approach. Optimizing $D$ to completion in the inner loop of training is computationally prohibitive, and on finite datasets would result in overfitting. Instead, we alternate between $k$ steps of optimizing $D$ and one step of optimizing $G$. This results in $D$ being maintained near its optimal solution, so long as $G$ changes slowly enough. The procedure is formally presented in Algorithm 1.

In practice, equation 1 may not provide sufficient gradient for $G$ to learn well. Early in learning, when $G$ is poor, $D$ can reject samples with high confidence because they are clearly different from the training data. In this case, $\log(1 - D(G(\boldsymbol{z})))$ saturates. Rather than training $G$ to minimize $\log(1 - D(G(\boldsymbol{z})))$ we can train $G$ to maximize $\log D(G(\boldsymbol{z}))$. This objective function results in the same fixed point of the dynamics of $G$ and $D$ but provides much stronger gradients early in learning.

## 4 Theoretical Results

The generator $G$ implicitly defines a probability distribution $p_g$ as the distribution of the samples $G(\boldsymbol{z})$ obtained when $\boldsymbol{z} \sim p_{\boldsymbol{z}}$. Therefore, we would like Algorithm 1 to converge to a good estimator of $p_{\text{data}}$, if given enough capacity and training time. The results of this section are done in a non-parametric setting, e.g. we represent a model with infinite capacity by studying convergence in the space of probability density functions.

We will show in section 4.1 that this minimax game has a global optimum for $p_g = p_{\text{data}}$. We will then show in section 4.2 that Algorithm 1 optimizes Eq 1, thus obtaining the desired result.

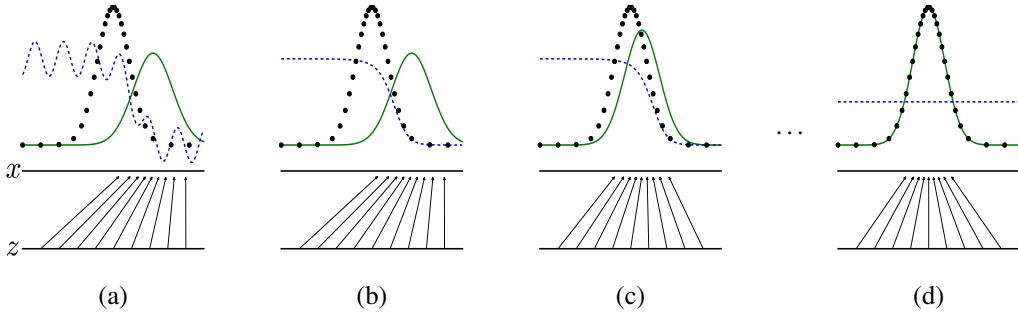

|       |       |       |       |
|-------|-------|-------|-------|
| (a)   | (b)   | (c)   | (d)   |

Figure 1: Generative adversarial nets are trained by simultaneously updating the **d**iscriminative distribution ($D$, blue, dashed line) so that it discriminates between samples from the data generating distribution (black, dotted line) $p_{\bm{x}}$ from those of the **g**enerative distribution $p_g$ (G) (green, solid line). The lower horizontal line is the domain from which $\bm{z}$ is sampled, in this case uniformly. The horizontal line above is part of the domain of $\bm{x}$. The upward arrows show how the mapping $\bm{x} = G(\bm{z})$ imposes the non-uniform distribution $p_g$ on transformed samples. $G$ contracts in regions of high density and expands in regions of low density of $p_g$. (a) Consider an adversarial pair near convergence: $p_g$ is similar to $p_{\text{data}}$ and $D$ is a partially accurate classifier. (b) In the inner loop of the algorithm $D$ is trained to discriminate samples from data, converging to $D^*(\bm{x}) = \frac{p_{\text{data}}(\bm{x})}{p_{\text{data}}(\bm{x}) + p_g(\bm{x})}$. (c) After an update to $G$, gradient of $D$ has guided $G(\bm{z})$ to flow to regions that are more likely to be classified as data. (d) After several steps of training, if $G$ and $D$ have enough capacity, they will reach a point at which both cannot improve because $p_g = p_{\text{data}}$. The discriminator is unable to differentiate between the two distributions, i.e. $D(\bm{x}) = \frac{1}{2}$.

---

**Algorithm 1** Minibatch stochastic gradient descent training of generative adversarial nets. The number of steps to apply to the discriminator, $k$, is a hyperparameter. We used $k = 1$, the least expensive option, in our experiments.

---

**for** number of training iterations **do**

    **for** $k$ steps **do**

        • Sample minibatch of $m$ noise samples $\{\bm{z}^{(1)}, \ldots, \bm{z}^{(m)}\}$ from noise prior $p_g(\bm{z})$.

        • Sample minibatch of $m$ examples $\{\bm{x}^{(1)}, \ldots, \bm{x}^{(m)}\}$ from data generating distribution $p_{\text{data}}(\bm{x})$.

        • Update the discriminator by ascending its stochastic gradient:

$$\nabla_{\theta_d} \frac{1}{m} \sum_{i=1}^{m} \left[ \log D\left(\bm{x}^{(i)}\right) + \log\left(1 - D\left(G\left(\bm{z}^{(i)}\right)\right)\right) \right].$$

    **end for**

    • Sample minibatch of $m$ noise samples $\{\bm{z}^{(1)}, \ldots, \bm{z}^{(m)}\}$ from noise prior $p_g(\bm{z})$.

    • Update the generator by descending its stochastic gradient:

$$\nabla_{\theta_g} \frac{1}{m} \sum_{i=1}^{m} \log\left(1 - D\left(G\left(\bm{z}^{(i)}\right)\right)\right).$$

**end for**

The gradient-based updates can use any standard gradient-based learning rule. We used momentum in our experiments.

---

## 4.1 Global Optimality of $p_g = p_{\text{data}}$

We first consider the optimal discriminator $D$ for any given generator $G$.

**Proposition 1.** *For $G$ fixed, the optimal discriminator $D$ is*

$$D_G^*(\bm{x}) = \frac{p_{data}(\bm{x})}{p_{data}(\bm{x}) + p_g(\bm{x})} \tag{2}$$

*Proof.* The training criterion for the discriminator D, given any generator $G$, is to maximize the quantity $V(G, D)$

$$V(G, D) = \int_{\boldsymbol{x}} p_{\text{data}}(\boldsymbol{x}) \log(D(\boldsymbol{x})) dx + \int_z p_{\boldsymbol{z}}(\boldsymbol{z}) \log(1 - D(g(\boldsymbol{z}))) dz$$

$$= \int_{\boldsymbol{x}} p_{\text{data}}(\boldsymbol{x}) \log(D(\boldsymbol{x})) + p_g(\boldsymbol{x}) \log(1 - D(\boldsymbol{x})) dx \tag{3}$$

For any $(a, b) \in \mathbb{R}^2 \setminus \{0, 0\}$, the function $y \to a \log(y) + b \log(1 - y)$ achieves its maximum in $[0, 1]$ at $\frac{a}{a+b}$. The discriminator does not need to be defined outside of $Supp(p_{\text{data}}) \cup Supp(p_g)$, concluding the proof. $\qquad \square$

Note that the training objective for $D$ can be interpreted as maximizing the log-likelihood for estimating the conditional probability $P(Y = y|\boldsymbol{x})$, where $Y$ indicates whether $\boldsymbol{x}$ comes from $p_{\text{data}}$ (with $y = 1$) or from $p_g$ (with $y = 0$). The minimax game in Eq. 1 can now be reformulated as:

$$
\begin{aligned}
C(G) &= \max_D V(G, D) \\
&= \mathbb{E}_{\boldsymbol{x} \sim p_{\text{data}}}[\log D_G^*(\boldsymbol{x})] + \mathbb{E}_{\boldsymbol{z} \sim p_{\boldsymbol{z}}}[\log(1 - D_G^*(G(\boldsymbol{z})))] \\
&= \mathbb{E}_{\boldsymbol{x} \sim p_{\text{data}}}[\log D_G^*(\boldsymbol{x})] + \mathbb{E}_{\boldsymbol{x} \sim p_g}[\log(1 - D_G^*(\boldsymbol{x}))] \\
&= \mathbb{E}_{\boldsymbol{x} \sim p_{\text{data}}}\left[\log \frac{p_{\text{data}}(\boldsymbol{x})}{P_{\text{data}}(\boldsymbol{x}) + p_g(\boldsymbol{x})}\right] + \mathbb{E}_{\boldsymbol{x} \sim p_g}\left[\log \frac{p_g(\boldsymbol{x})}{p_{\text{data}}(\boldsymbol{x}) + p_g(\boldsymbol{x})}\right]
\end{aligned}
\tag{4}
$$

**Theorem 1.** *The global minimum of the virtual training criterion $C(G)$ is achieved if and only if $p_g = p_{data}$. At that point, $C(G)$ achieves the value $-\log 4$.*

*Proof.* For $p_g = p_{\text{data}}$, $D_G^*(\boldsymbol{x}) = \frac{1}{2}$, (consider Eq. 2). Hence, by inspecting Eq. 4 at $D_G^*(\boldsymbol{x}) = \frac{1}{2}$, we find $C(G) = \log \frac{1}{2} + \log \frac{1}{2} = -\log 4$. To see that this is the best possible value of $C(G)$, reached only for $p_g = p_{\text{data}}$, observe that

$$\mathbb{E}_{\boldsymbol{x} \sim p_{\text{data}}}[-\log 2] + \mathbb{E}_{\boldsymbol{x} \sim p_g}[-\log 2] = -\log 4$$

and that by subtracting this expression from $C(G) = V(D_G^*, G)$, we obtain:

$$C(G) = -\log(4) + KL\left(p_{\text{data}} \,\middle\|\, \frac{p_{\text{data}} + p_g}{2}\right) + KL\left(p_g \,\middle\|\, \frac{p_{\text{data}} + p_g}{2}\right) \tag{5}$$

where KL is the Kullback–Leibler divergence. We recognize in the previous expression the Jensen–Shannon divergence between the model's distribution and the data generating process:

$$C(G) = -\log(4) + 2 \cdot JSD\left(p_{\text{data}} \,\|\, p_g\right) \tag{6}$$

Since the Jensen–Shannon divergence between two distributions is always non-negative, and zero iff they are equal, we have shown that $C^* = -\log(4)$ is the global minimum of $C(G)$ and that the only solution is $p_g = p_{\text{data}}$, i.e., the generative model perfectly replicating the data distribution. $\quad \square$

## 4.2 Convergence of Algorithm 1

**Proposition 2.** *If $G$ and $D$ have enough capacity, and at each step of Algorithm 1, the discriminator is allowed to reach its optimum given $G$, and $p_g$ is updated so as to improve the criterion*

$$\mathbb{E}_{\boldsymbol{x} \sim p_{data}}[\log D_G^*(\boldsymbol{x})] + \mathbb{E}_{\boldsymbol{x} \sim p_g}[\log(1 - D_G^*(\boldsymbol{x}))]$$

*then $p_g$ converges to $p_{data}$*

*Proof.* Consider $V(G, D) = U(p_g, D)$ as a function of $p_g$ as done in the above criterion. Note that $U(p_g, D)$ is convex in $p_g$. The subderivatives of a supremum of convex functions include the derivative of the function at the point where the maximum is attained. In other words, if $f(x) = \sup_{\alpha \in \mathcal{A}} f_\alpha(x)$ and $f_\alpha(x)$ is convex in $x$ for every $\alpha$, then $\partial f_\beta(x) \in \partial f$ if $\beta = \arg \sup_{\alpha \in \mathcal{A}} f_\alpha(x)$. This is equivalent to computing a gradient descent update for $p_g$ at the optimal $D$ given the corresponding $G$. $\sup_D U(p_g, D)$ is convex in $p_g$ with a unique global optima as proven in Thm 1, therefore with sufficiently small updates of $p_g$, $p_g$ converges to $p_x$, concluding the proof. $\quad \square$

In practice, adversarial nets represent a limited family of $p_g$ distributions via the function $G(\boldsymbol{z}; \theta_g)$, and we optimize $\theta_g$ rather than $p_g$ itself, so the proofs do not apply. However, the excellent performance of multilayer perceptrons in practice suggests that they are a reasonable model to use despite their lack of theoretical guarantees.

| Model | MNIST | TFD |
|---|---|---|
| DBN [3] | $138 \pm 2$ | $1909 \pm 66$ |
| Stacked CAE [3] | $121 \pm 1.6$ | $\mathbf{2110 \pm 50}$ |
| Deep GSN [5] | $214 \pm 1.1$ | $1890 \pm 29$ |
| Adversarial nets | $\mathbf{225 \pm 2}$ | $\mathbf{2057 \pm 26}$ |

Table 1: Parzen window-based log-likelihood estimates. The reported numbers on MNIST are the mean log-likelihood of samples on test set, with the standard error of the mean computed across examples. On TFD, we computed the standard error across folds of the dataset, with a different $\sigma$ chosen using the validation set of each fold. On TFD, $\sigma$ was cross validated on each fold and mean log-likelihood on each fold were computed. For MNIST we compare against other models of the real-valued (rather than binary) version of dataset.

## 5 Experiments

We trained adversarial nets an a range of datasets including MNIST[21], the Toronto Face Database (TFD) [27], and CIFAR-10 [19]. The generator nets used a mixture of rectifier linear activations [17, 8] and sigmoid activations, while the discriminator net used maxout [9] activations. Dropout [16] was applied in training the discriminator net. While our theoretical framework permits the use of dropout and other noise at intermediate layers of the generator, we used noise as the input to only the bottommost layer of the generator network.

We estimate probability of the test set data under $p_g$ by fitting a Gaussian Parzen window to the samples generated with $G$ and reporting the log-likelihood under this distribution. The $\sigma$ parameter of the Gaussians was obtained by cross validation on the validation set. This procedure was introduced in Breuleux *et al.* [7] and used for various generative models for which the exact likelihood is not tractable [24, 3, 4]. Results are reported in Table 1. This method of estimating the likelihood has somewhat high variance and does not perform well in high dimensional spaces but it is the best method available to our knowledge. Advances in generative models that can sample but not estimate likelihood directly motivate further research into how to evaluate such models. In Figures 2 and 3 we show samples drawn from the generator net after training. While we make no claim that these samples are better than samples generated by existing methods, we believe that these samples are at least competitive with the better generative models in the literature and highlight the potential of the adversarial framework.

## 6 Advantages and disadvantages

This new framework comes with advantages and disadvantages relative to previous modeling frameworks. The disadvantages are primarily that there is no explicit representation of $p_g(\boldsymbol{x})$, and that $D$ must be synchronized well with $G$ during training (in particular, $G$ must not be trained too much without updating $D$, in order to avoid "the Helvetica scenario" in which $G$ collapses too many values of $\mathbf{z}$ to the same value of $\mathbf{x}$ to have enough diversity to model $p_{\text{data}}$), much as the negative chains of a Boltzmann machine must be kept up to date between learning steps. The advantages are that Markov chains are never needed, only backprop is used to obtain gradients, no inference is needed during learning, and a wide variety of functions can be incorporated into the model. Table 2 summarizes the comparison of generative adversarial nets with other generative modeling approaches.

The aforementioned advantages are primarily computational. Adversarial models may also gain some statistical advantage from the generator network not being updated directly with data examples, but only with gradients flowing through the discriminator. This means that components of the input are not copied directly into the generator's parameters. Another advantage of adversarial networks is that they can represent very sharp, even degenerate distributions, while methods based on Markov chains require that the distribution be somewhat blurry in order for the chains to be able to mix between modes.

## 7 Conclusions and future work

This framework admits many straightforward extensions:

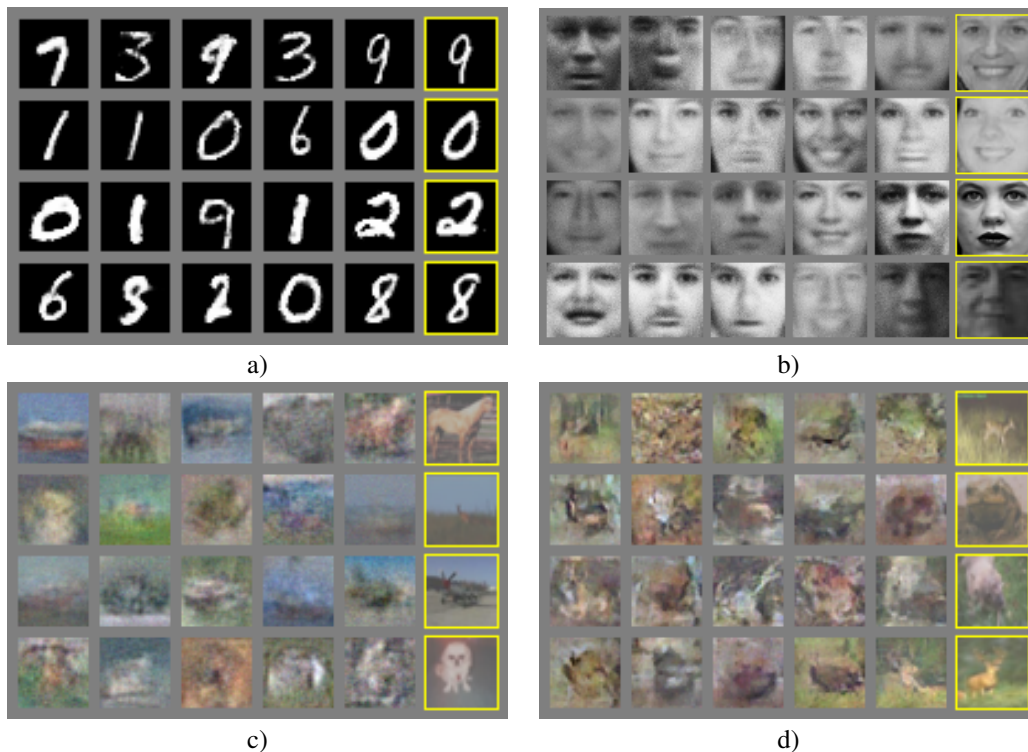

Figure 2: Visualization of samples from the model. Rightmost column shows the nearest training example of the neighboring sample, in order to demonstrate that the model has not memorized the training set. Samples are fair random draws, not cherry-picked. Unlike most other visualizations of deep generative models, these images show actual samples from the model distributions, not conditional means given samples of hidden units. Moreover, these samples are uncorrelated because the sampling process does not depend on Markov chain mixing. a) MNIST b) TFD c) CIFAR-10 (fully connected model) d) CIFAR-10 (convolutional discriminator and "deconvolutional" generator)

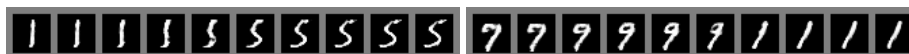

Figure 3: Digits obtained by linearly interpolating between coordinates in $z$ space of the full model.

1. A *conditional generative* model $p(\boldsymbol{x} \mid \boldsymbol{c})$ can be obtained by adding $\boldsymbol{c}$ as input to both $G$ and $D$.
2. *Learned approximate inference* can be performed by training an auxiliary network to predict $\boldsymbol{z}$ given $\boldsymbol{x}$. This is similar to the inference net trained by the wake-sleep algorithm [15] but with the advantage that the inference net may be trained for a fixed generator net after the generator net has finished training.
3. One can approximately model all conditionals $p(\boldsymbol{x}_S \mid \boldsymbol{x}_{\not S})$ where $S$ is a subset of the indices of $\boldsymbol{x}$ by training a family of conditional models that share parameters. Essentially, one can use adversarial nets to implement a stochastic extension of the deterministic MP-DBM [10].
4. *Semi-supervised learning*: features from the discriminator or inference net could improve performance of classifiers when limited labeled data is available.
5. *Efficiency improvements:* training could be accelerated greatly by devising better methods for coordinating $G$ and $D$ or determining better distributions to sample $\mathbf{z}$ from during training.

This paper has demonstrated the viability of the adversarial modeling framework, suggesting that these research directions could prove useful.

|  | Deep directed graphical models | Deep undirected graphical models | Generative autoencoders | Adversarial models |
|---|---|---|---|---|
| Training | Inference needed during training. | Inference needed during training. MCMC needed to approximate partition function gradient. | Enforced tradeoff between mixing and power of reconstruction generation | Synchronizing the discriminator with the generator. Helvetica. |
| Inference | Learned approximate inference | Variational inference | MCMC-based inference | Learned approximate inference |
| Sampling | No difficulties | Requires Markov chain | Requires Markov chain | No difficulties |
| Evaluating $p(x)$ | Intractable, may be approximated with AIS | Intractable, may be approximated with AIS | Not explicitly represented, may be approximated with Parzen density estimation | Not explicitly represented, may be approximated with Parzen density estimation |
| Model design | Models need to be designed to work with the desired inference scheme — some inference schemes support similar model families as GANs | Careful design needed to ensure multiple properties | Any differentiable function is theoretically permitted | Any differentiable function is theoretically permitted |

Table 2: Challenges in generative modeling: a summary of the difficulties encountered by different approaches to deep generative modeling for each of the major operations involving a model.

## Acknowledgments

We would like to acknowledge Patrice Marcotte, Olivier Delalleau, Kyunghyun Cho, Guillaume Alain and Jason Yosinski for helpful discussions. Yann Dauphin shared his Parzen window evaluation code with us. We would like to thank the developers of Pylearn2 [11] and Theano [6, 1], particularly Frédéric Bastien who rushed a Theano feature specifically to benefit this project. Arnaud Bergeron provided much-needed support with LaTeX typesetting. We would also like to thank CIFAR, and Canada Research Chairs for funding, and Compute Canada, and Calcul Québec for providing computational resources. Ian Goodfellow is supported by the 2013 Google Fellowship in Deep Learning. Finally, we would like to thank Les Trois Brasseurs for stimulating our creativity.

## Footnotes

*Ian Goodfellow is now a research scientist at Google, but did this work earlier as a UdeM student

†Jean Pouget-Abadie did this work while visiting Université de Montréal from Ecole Polytechnique.

‡Sherjil Ozair is visiting Université de Montréal from Indian Institute of Technology Delhi

§Yoshua Bengio is a CIFAR Senior Fellow.

[1]All code and hyperparameters available at `http://www.github.com/goodfeli/adversarial`

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
