[Reviews · NeurIPS 2014]

Submitted by Assigned_Reviewer_16

Line 33: I don't think it is accurate to attribute the recent success of supervised neural nets on various applications to BP and dropout. Firstly, learning nets with gradient descent has been around a long time, and the key to its recent success has mostly been fast computers/GPUs, a wealth of labelled data, advances in understanding of how to make SGD work well (e.g. good random initialization), and the use of dataset dependent priors like convolutions to mitigate the overfitting problems. Techniques like dropout have also been useful in reducing overfitting, but are hardly the key missing ingredient to make these systems work well.

Line 37: The claim that the lacklustre the results associated with unsupervised generative approaches is owed purely to their intractability issues is a strong and problematic one. Even if we had perfect inference and exact maximum likelihood learning in these models, it is not clear at all that they would suddenly be very useful for solving tasks like image labelling.

Line 50: Dropout isn't a training algorithm. It is a regularization method, and implies a different model. Also, backpropagation isn't a training method so much as it is a method for computing gradients. SGD is the algorithm typically used to training neural nets in this context.

Line 58: What is a "parameterized generative machine"?

Line 195, 219: The proofs of these very simple propositions don't really themselves provide insight into what is going on, so they can safely be put in the appendix. I think the question one should ask when deciding if a proof should go in the appendix is: does the *proof itself* (not the result) provide some new idea that deepens ones understanding of what is going on, or are they just routine computations?

Line 247: This proposition is problematic. Even if G and D have "enough capacity", they may still be parameterized in such a way that the resulting optimization is not convex/concave in either of their parameters.

In general, for powerful models of the kind used in practice, they will have such a parameterization.

Also, even if we do optimization directly in function space (which is impossible in practice), we would need that the function space is itself a convex set (such as the space of all functions). If not, a hypothetical local optimizer over functions in the function space could easily run into problems, even if the objective function is convex/concave.

Thus this kind of result only holds if the functions can themselves be optimized in an unrestricted function space, which isn't explicitly mentioned.

I don't know of any good parametric models which give a convex function sub-space and a parameterization which would allow this kind of optimization to be convex/concave, so basically this result says nothing about realistic situations.

line 367: what do you mean by "explicit representation" here?

Line 428: The same paper appears twice in the bibliography.
Summary: This paper introduces a scheme for training generative models where the goal is to fool a discriminator system that tries to separate data from the true distribution from data generated by the model.

The idea is interesting and seemingly novel, and the paper is mostly well written. I think it should be accepted.

That said, I wish the authors had developed the paper more, as it feels quite preliminary. The theoretical work is primitive, and the experiments are pretty basic. The paper feels a bit padded out with an overlong review of other generative modeling approaches, and simple computational/technical proofs taking up space in main body.

Submitted by Assigned_Reviewer_19

Review of submission 1384:
Generative Adversarial Nets

Summary:

There are two "adversarial" MLPs G and D. G gets random inputs, and should learn to generate samples of a given data distribution. D is trained to learn the probability that something like G's output stems from the original distribution instead of G, while G is trained "to maximize the probability of D making a mistake." First experimental results are described.

Comments:

Sections 1 and 2:

I find the basic ideas quite interesting and appealing.

The discussion of related work, however, could be a bit more balanced. Example:

"So far, the most striking successes in deep learning have involved discriminative models, usually those that map a high-dimensional, rich sensory input to a class label [14, 21]."

References [14, 21] are about discriminative models for speech recognition and Imagenet. But [14] does not yet include the current state of the art of discriminative models for speech - compare instead the work of Graves et al, e.g., IJCAI 2007, ICASSP 2013, ICML 2014. And [21] is not any longer state of the art for Imagenet - compare instead work of Zeiler and Fergus http://arxiv.org/abs/1311.2901

Also, the cited survey of 2009 [1] on "rich, hierarchical models [1]" does not mention numerous original papers on deep learning with hierarchical models since the 1960s, many of them listed in a recent comprehensive overview http://arxiv.org/abs/1404.7828

In the context of "generalized denoising auto-encoders [4]", one should also cite the earlier, closely related work of Behnke at IJCAI 2001 and ICANN 2003.

In the context of backpropagation "into a stochastic network with continuous activations (so that gradients can be back-propagated)," one should also cite the much older original papers on this, in particular those by Williams (e.g., Machine Learning, 1992), who adjusted mean and variance of probabilistic continuous units by backpropagation through random number generators.

Finally, how is the submission related to the first work on "adversarial" MLPs for modeling data distributions through estimating conditional probabilities, which was called "predictability minimisation" or PM (Schmidhuber, NECO 1992)? The new approach seems similar in many ways. Both approaches use "adversarial" MLPs to estimate certain probabilities and to learn to encode distributions. A difference is that the new system learns to generate a non-trivial distribution in response to statistically independent, random inputs, while good old PM learns to generate statistically independent, random outputs in response to a non-trivial distribution (by extracting mutually independent, factorial features encoding the distribution). Hence the new system essentially inverts the direction of PM - is this the main difference? Should it perhaps be called "inverse PM"?

Sections 4 and 6 - convergence:

Interesting theorem: Under certain assumptions the algorithm will converge:

"If G and D have enough capacity, and at each step of Algorithm 1, D is allowed to approximately track its optimum given G, and each update of G improves its criterion, then Algorithm 1 converges to the global optimum."

Here I seem to stumble about a possibly irrelevant or trivial issue. Is it really enough that each update of G improves its criterion? Could it not be that those improvements get exponentially smaller over time in some weird way, such that even infinitely many of them might converge to a suboptimal point? My intuition tells me that this is not the case, and probably this is just a minor technical issue, but has this really been shown, or is it trivial for some reason - if so, is there a reference for this?

As was to be expected, some of the drawbacks mentioned by the authors are similar to those of adversarial PM for distribution modeling: "G must not be trained too much without updating D ..."

Section 5 - experiments

Experimental results and graphical visualizations are interesting.

They seem a bit hard to judge tough, for lack of direct comparisons to the state of the art, although at least for one of the experiments (Parzen density) the authors cautiously write: "This method of estimating the likelihood has somewhat high variance and does not perform well in high dimensional spaces but it is the best method available to our knowledge."

The authors comment on the experimental results with some handwaving:

"While we make no claim that these samples are better than samples generated by existing methods, we believe that these samples are at least competitive with the better generative models in the literature and highlight the potential of the adversarial framework."

Section 7 - a somewhat half-hearted conclusion:

"This paper has demonstrated the viability of the adversarial modeling framework, suggesting that these research directions could prove useful."

Summary: The authors present an interesting way of using adversarial nets to model distributions. The approach seems to invert the direction of predictability minimization, an adversarial technique for distribution modeling of the 1990s. The discussion of relations to previous work still leaves something to be desired. The experimental results (although interesting) do not yet quite convincingly show that the approach can significantly improve data distribution recovery (and deep learning in general). In the present form the submission may not yet be quite strong enough for NIPS. But the authors should be encouraged to pursue this further, perhaps for the next ICML or NIPS?

Submitted by Assigned_Reviewer_32

This paper proposes a new framework to train a generative model via backpropagation and without sampling.
Interesting samples are shown and the likelihood is competitive. The training procedure sounds hard to get right but it is an interesting step in the
direction of fast generative models. More results of how this helps on real tasks or real datasets would have made the paper significantly stronger.
Another interesting endevaour could be to train this on language to generate real sentences that make sense.

The caption part of Fig.1: "(a) Consider an adversarial pair near convergence" is not very intuitive. Shouldn't it be *not* yet near convergence?

typos:
- algorithms for many kinds of model and optimization algorithm
- trained adversarial nets an a range of
- Parzen widow-based log-likelihood
Summary: This paper proposes a new framework to train a generative model via backpropagation and without sampling.
Interesting samples are shown and the likelihood is competitive. The training procedure sounds hard to get right but it is an interesting step in the
direction of fast generative models. More results of how this helps on real tasks or real datasets would have made the paper significantly stronger.
Another interesting endevaour could be to train this on language to generate real sentences that make sense.

Author Feedback
Author rebuttal: R16: Why should generative models be good for image labeling?
We do not claim that generative models should be good for image labeling, just that they could be
useful. Possible applications involve speech recognition and machine translation, where a good
language model is necessary, or speech synthesis, where the goal of the system is to produce high
quality samples.

R19 / R32: It should be evaluated on more / more "real" datasets before it can be accepted to NIPS
Compare other recent papers on generative models at NIPS/ICML:
ICML 2014 Deep Autoregressive Networks: evaluated on MNIST, synthetic data, and very small UCI datasets
ICML 2014 Deep Generative Stochastic Networks Trainable by Backprop: evaluted on MNIST and TFD
ICML 2014 Neural Variational Inference and Learning in Belief Networks: evaluated on MNIST and 20 Newsgroups
NIPS 2013 Multi-Prediction Deep Boltzman Machines: evaluated on MNIST and NORB
Among these, TFD is the perhaps the most advanced dataset evaluated.
We also evaluated TFD, putting us on the same footing as other research in this field,
but also introduce results on CIFAR-10.

R32: You should use it to generate sentences
We agree this is an interesting direction, but will require an extension to discrete observed
variables. We thus think it is best to leave to future work.

R19: How is this related to PM?
PM regularizes the hidden units of a generative model to be statistically independent from each other.
GANs express no preference about how the hidden units are distributed. We used independent priors in
the experiments for this paper, but that's not an essential part of the approach. PM involves competition
between two MLPs but the idea of competition is more qualitative than formal--each network has its own
objective function qualitatively designed to make them compete, while GANs play a minimax game on a single
value function. Nearly all forms of engineering involve competition between two forces of some kind,
whether it is mechanical systems exploiting Newton's 3rd law or Boltzmann machines inducing competition
between the positive and negative phase of learning.

R16: "The paper feels a bit padded out with an overlong review of other generative modeling approaches"
It's difficult to write a paper in this area without multiple authors writing to you asking you to cite their
paper. We've had numerous authors write to us asking us to cite their generative modeling papers
in addition to the ones we already cite. The final version of the paper will probably still have a long
review section, in order to accommodate the citations requested by R19 as well as people who have
contacted us personally.

R16: The optimization problem in parameter space is not convex.
(*) Yes, our proof is in function space, to show that the criterion is sensible. Our proof is the game theory
analog of the way that objective function-based criteria such as score matching, noise contrastive estimation,
etc. are proven ``asymptotically consistent.'' These criteria are recognized as theoretically consistent because
the data distribution is a global maximum of the criterion, even though optimization of any actual parametric
distribution is likely to get stuck in a local minimum or plateau. We can add more explanation of how
optimization in function space is not feasible in practice and what problems can result.

R16: Supervised nets are doing well because of large datasets and GPUs, not backprop.
We mean that backprop allows computation of the exact gradient. Learning algorithms that rely
on MCMC approximations of the gradient have not been able to benefit much from the GPU revolution
because MCMC approximations get too inaccurate for large models. We can change the paper to
clarify this.

R16: How do you know the problem with generative models is optimization?
It's generally accepted that deep Boltzmann machines can't fit the training set very well. If you
turn up the number of Markov chain transitions the training set error always goes down.

R16: Dropout isn't that important to recent successes.
We can remove the discussion of dropout from the introduction.

R19: Convergence. In the consistency analysis setup (see (*) above) for
the *criterion*, we consider optimization in a (convex) function space,
and here it yields a min-max formulation common in game theory, which
is guaranteed to have a single solution because of the
convexity/concavity of the criterion. General conditions for
convergence of gradient-based optimization are reviewed for example in
(Bottou 2004, "Stochastic Learning"). With regular gradient steps (not
stochastic), it is enough that the learning rate be below some critical
value related to the maximum curvature. With stochastic gradient, a 1/t
decrease rate suffices. Under the assumption that D tracks G, it's
like if we are only optimizing the criterion for G, but with D
optimized in an inner loop.